# Tunable Multiple Surface Plasmonic Bending Beams into Single One by Changing Incident Light Wavelength

**Hang Zhang** [1,†], **Liang Wang** [1,†], **Xueli Li** [1], **Xiaoming Li** [1] **and Hui Li** [1,2,*]

1   School of Physical Science and Technology, Kunming University, Kunming 650214, China; zh56802022@126.com (H.Z.); liangwang122022@126.com (L.W.); lixueli2022@126.com (X.L.); lixiaoming11232022@126.com (X.L.)
2   Key Laboratory of Artificial Microstructures in Yunnan Higher Education Institutions, Kunming University, Kunming 650214, China
*   Correspondence: huili009@126.com
†   These authors contributed equally to this work.

**Abstract:** Controllable surface plasmonic bending beams (SPBs) with propagating along bending curves have a wide range of applications in the fields of fiber sensors, optical trapping, and micro-nano manipulations. In terms of designing and optimizing controllable SPB generators, there is great significance in realizing conversion between multiple SPBs and single SPB without rebuilding metasurface structures. In this study, a SPB generator, composed of an X-shaped nanohole array, is proposed to realize conversion between multiple SPBs and a single one by changing the incident light wavelength. The Fabry–Pérot (F–P) resonance effect of SPPs in nanoholes and localized surface plasmonic (LSP) resonance of the nanohole are utilized to explain this conversion. It turns out that the relationship between the electric field intensities of SPBs and the polarization angle of incident light satisfies the sine distribution, which is consistent with dipole radiation theory. In addition, we also find that the electric field intensities of SPBs rely on the width, length, and angle of the X-shaped nanohole. These findings could help in designing and optimizing controllable and multi-functions SPBs converters.

**Keywords:** surface plasmon polaritons; surface plasmonic bending beams; phase modulation

## 1. Introduction

Surface plasmonic polaritons (SPPs), special electromagnetic waves in a two-dimensional system consisting of collective electron oscillations and propagating on the metal interface, have important applications in the fields of photonics and electronics [1–3]. By artificially regulating SPPs, it is possible to create surface plasmonic bending beams (SPBs), which maintain their forms while traveling through arbitrary bending curves [4]. Due to arbitrary self-accelerating properties of SPBs waves, they have potential applications in sensors [5–10], optical trapping [11], and photon manipulations [12–14]. Various SPBs, including focused SPPs [15], Airy beams [16–18], Arbitrary bending beams [4,19,20], Weber beams, and Mathieu beams [21], have been realized by various artificial microstructures (such as hole array [4,17,22], grains array [19], two-dimensional binary phase mask gratings [4,19], and polymethyl methacrylate microsphere [23,24]). Dynamic focusing SPPs [25,26] were also realized by circular vertical slit array [27] and circular cross slit array [28]. These nanostructures, as polarization-sensitive nanostructures, were designed by utilizing phase modulation method [20], and can independently control phase and amplitude of SPPs by controlling parameters, size, and arrangement of structures [23–29]. However, their size is very small, and it is still a challenge to simultaneously generate the randomly multiplexed polarization states.

Recently, metasurfaces, two-dimensional (2D) artificial structures composed of arrays of subwavelength-size unit cells, have been widely used for wavefront reshaping [30–34].

Compared to traditional three-dimensional (3D) metamaterials, they can deeply interact with light and exploit new degrees of freedom to manipulate optical fields. Various physical effects, including the photonic Spin Hall effect [30,31,35,36], the momentum-space polarization effect [30], the interplay effect of the Pancharatnam–Berry (P-B) phase and Spin-orbit, and the interplay effect of the geometric phase and the dynamic phase [37,38], were applied to realize various SPBs by designing various metasurface structures. These SPBs include focused SPPs [39], orbital angular momentum (OAM) beams [40–43], arbitrary vector beams [44], Airy beams [45,46], and Bessel beams [37]. For example, a single ultrathin metasurface, composed of eight groups of rectangular nanoparticle arrays with various azimuth angles, was used to generate arbitrary vector beams [47]. An assembly of circularly polarized (CP) and linearly polarized (LP) states can be simultaneously generated by a metasurface made of L-shaped resonators with different geometrical features [48]. The polarization states and propagation direction of the desired output SPB can be accurately tuned by selecting the geometrical shape, size, and spatial sequence of each resonator in the unit cell [43–48]. By changing the orientation angle of hyperbolic metamaterials (HMMs) unit, the local amplitude and phase distributions of the transmitted electromagnetic waves passing through such HMMs unit can be adjusted to follow the Airy function within a wide spectral range (generation of broadband Airy beams) [49]. With a well-designed array of deep sub-wavelength nanostructures, separating photon information can be multiplexed into different channels or combined with different functions [50]. For the purpose of focusing two separated points in the predetermined plane, a spin-selected metasurface lens was created [51,52]. The broadband, high-efficiency, and high-quality polarization-controlled self-accelerating beam were achieved by designing versatile dielectric metasurfaces [53–56]. Novel physical models and artificial metasurfaces have been utilized to increase the depth and scope of the research on control SPBs. However, there is not enough discussion about converting multiple SPBs into one without rebuilding structures.

In this study, the phase modulation method was utilized to design an X-shaped nanohole array and multiple SPBs were generated. The angle between multiple SPBs was dynamically controlled by changing the incident light wavelength. A conversion between multiple SPBs and single one was realized by only changing the incident light wavelength. We applied the Fabry–Pérot (F–P) resonance effect of SPPs in the nanohole and to the localized surface plasmonic (LSP) resonance of the nanohole to explain this conversion. The electric field intensities of these SPBs were controlled by the polarization angle of input light waves. The relationship between the electric field intensities of the SPBs and the polarization angle of incident light satisfied the sine distribution, consistent with theoretical analysis of dipole radiation. The effects of the structural parameters on the SPBs were investigated. The results show that the electric field intensities of SPBs rely on the structural width and length. Compared to the effect of length, the impact of width and angle between the two arms of an X-shaped nanohole on the electric field intensities of SPBs is more obvious. These findings could help in designing and optimizing controllable bending beam generators.

## 2. Theoretical Analysis and Structure

SPBs, as an electromagnetic wave propagating in the two-dimensional (x-y) plane, can be expressed in the Helmholtz equation [9,17,19,20],

$$\nabla^2 E_{spp}(x,y) + k_{spp}^2 E_{spp}(x,y) = 0, \tag{1}$$

where $k_{spp}$ represents the SPPs wave vector. According to the Huygens–Fresnel principle, the superimposed SPPs field generated by a single nanohole at any position P($x$, $y$) is written as,

$$E_n(x,y) = \frac{1}{i\lambda} \int E_0(x_0, y_0) \frac{K(\theta_n)}{r_n} e^{-(i\vec{k}_{spp}\vec{r}_n)} ds, \tag{2}$$

where $r_n$ represents the distance between each point on a wavefront. $K(\theta_n)$ is a direction factor. $\lambda$ is the incident light wavelength. $E_0(x_0, y_0)$ represents the electric field distribution by a single nanohole. If only the second order is taken into consideration, $r_n$ can be expressed as follows using the binomial expansions,

$$
\begin{aligned}
r_n &= \sqrt{z^2 + (x - x_0)^2 + (y - y_0)^2} \\
&\approx z\left\{1 + \frac{1}{2}\left[\left(\frac{x-x_0}{z}\right)^2 + \left(\frac{y-y_0}{z}\right)^2\right] - \frac{1}{8}\left[\left(\frac{x-x_0}{z}\right)^4 + \left(\frac{y-y_0}{z}\right)^4\right]\right\}.
\end{aligned} \tag{3}
$$

If we consider $r_n^2 \approx z^2$ and $K(\theta_n) \approx \cos(\theta_n) \approx \frac{z}{r_n}$, we set $M = (x - x_0)^2 + (y - y_0)^2$ and $Q = (x - x_0)^4 + (y - y_0)^4$. Equation (2) is then expressed as,

$$
E_n(x, y) \approx \frac{1}{i\lambda z} \int E_0(x_0, y_0) e^{-ik_{spp}\left[z + \frac{1}{2z}M - \frac{1}{8z^3}Q\right]} ds. \tag{4}
$$

Obviously, according to the superposition of electromagnetic waves, $E_{spp}(x, y) = \sum E_n(x, y)$ can be written as,

$$
\begin{aligned}
E_{spp} \quad (x, y) &\propto \frac{N}{i\lambda z} E_0(x_0, y_0) e^{-ik_{spp}r_1}\left(1 + \sum e^{ik_{spp}\Delta r_n}\right) \\
&\propto \frac{N}{i\lambda z} E_0' \sin\beta[1 + (c_1/c_2)\cot\beta]\sum e^{-ik_{spp}z}\sum e^{-ik_{spp}\frac{M}{2z}}\sum e^{-ik_{spp}\frac{Q}{8z^3}}
\end{aligned} \tag{5}
$$

where $c_1$ and $c_2$ are the SPPs coupling coefficients with $x$ and $y$ polarization, respectively, and $\beta$ is the polarization angle of incident light waves. $N$ is the number of sources $\Delta r_n = r_{n+1} - r_n$, $\varphi(x) = \sum \varphi_n(x) = \sum k_{spp}\Delta r_n$ represents the phase, and $k_{spp} = \frac{2\pi}{\lambda_{spp}}$ represents the SPPs wave vector. Equation (5) explains theoretically that both the amplitude and phase of the SPPs waves can be controlled by $\beta$, $c_1$, $c_2$, and $\varphi(x)$. The phase $\varphi(x)$ mainly depends on the arrangement of the structure array. As a result, distinct SPBs propagation trajectories with fixed $c_1$ and $c_2$ correspond to different phase $\varphi(x)$ distributions. The different angle $\beta$ corresponds to different $E_{spp}(x, y)$ with fixed $c_1$, $c_2$, and $\varphi(x)$, and results in different electric field intensities distribution. In addition, shown in Equation (5), the $\varphi(x)$ is expressed as,

$$
\varphi(x) = \sum k_{spp}\Delta r_n \approx \varphi_1(x) + \varphi_2(x) \approx \sum k_{spp}\frac{M}{2z} + \sum k_{spp}\frac{Q}{8z^3}. \tag{6}
$$

Theoretically, if the $E_0(x_0, y_0)$ is large enough, ($|E_0(x_0, y_0)\varphi_1(x)| >> |E_0(x_0, y_0)\varphi_2(x)|$), just the first term holds true in Equation (5), and a single SPB can be generated as a result. If the $E_0(x_0, y_0)$ is weak, ($|E_0(x_0, y_0)\varphi_1(x)| \approx |E_0(x_0, y_0)\varphi_2(x)|$), the first two terms at least are acceptable in Equation (5), and as a result, multiple SPBs can be generated at once.

In this study, to achieve SPBs, the phase modulation method was utilized [9,17,19,20,22]. Under the non-paraxial regimes, $\sin\theta = \tan\theta/\sqrt{1 + \tan^2\theta}$, the phase of required SPBs can be obtained, $\phi(x) = -\int k_0 \tan\theta/\sqrt{1 + \tan^2\theta}dx$, where $\tan\theta = f'(y)$, and $f'(y)$ is the first-order derivative of the designed bending trajectory $f(y)$ [20,50,57,58]. For simplicity, the paraxial regime ($\sin\theta \approx \tan\theta$) was also applied, and the required phase can be obtained by $\phi(x) = -\int k_0 \sin\theta dx$. In this study, the quadratic curve ($f(y) = -ay^2$) was chosen, where constant $a$ is $1.13 \times 10^{-2}$. According to equation $\phi(x) = -\int k_0 \sin\theta dx$, the desired phase ($\phi(x) = -1.33kax^{1.5}$) was obtained for paraxial regimes. The location of every X-shaped nanohole was calculated by solving the equation $\varphi(x) = \phi(x)$ [17,19,20,22,59]. The distances $H_n = y_{n+1} - y_n$ were calculated when the number of X-shaped nanohole was equal to 20, which is shown in Figure 1a. The corner symbol n represents number of radiation sources.

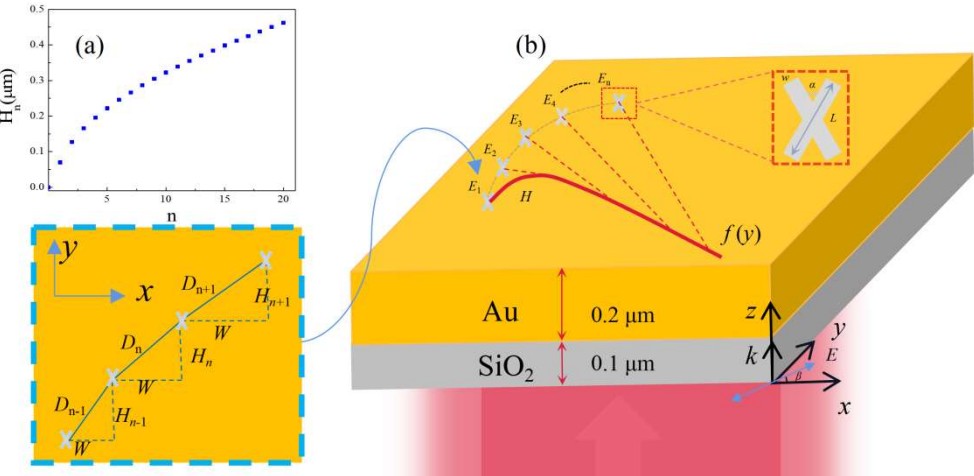

**Figure 1.** (**a**) Calculated distances $H_n$ of the radiation source. (**b**) Configuration of the Au/SiO$_2$ structure. The red block diagram illustrates a local amplification of the structure and structural parameter.

The configuration of an Au/SiO$_2$ structure is shown in Figure 1b. The thickness of the gold film and SiO$_2$ substrate were fixed at 0.2 μm and 0.1 μm, respectively. The X-shaped nanohole array (with the number of X-shaped nanohole equal to 20) was designed on the Au film. The inset is the single X-shaped nanohole in the red wireframe. The values given for $L$ and $w$, which stand for the arm length and width of an X-shaped nanohole, respectively, were 0.3 μm and 0.05 μm. The angle $\alpha$ is the angle between two arms. $D_n$ represents the distance between adjacent radiation sources, $\beta$ represents the polarization angle between the in-plane incident electric $E_0$ and the $x$ axis. $W$ and $H_n$ represent the distances in $x$ and $y$ direction between adjacent radiation sources, respectively. In this paper, the distance $W$ was set as 0.6 μm. The distance between adjacent radiation sources, $D_n = \sqrt{W^2 + (H_n)^2}$, is aperiodic. The finite-difference time-domain (FDTD) approach was used to model the SPPs propagation characteristics. Lumerical FDTD is 3D/2D maxwell's solver for nanophotonic devices, processes, and materials. The Au(gold)-CRC material model in Lumerical FDTD solutions software was selected. Boundary conditions for perfect matching layers (PML) were utilized. The frequency-domain field profile monitor was applied. The apodization time width was 100 fs. The mesh type of auto non-uniform was selected. The minimum mesh step of $2 \times 10^{-5}$ μm was applied. The total number of FDTD Yee nodes was 176.992 MNodes. A plane wave with black/periodic type was an incident along direction $z$ into the X-shaped nanohole array. For the calculations, we used an AMD Ryzen ThreadripperPRO 3945WX 12-Cores @4.00 GHz, with 128.0 GB installed memory. It took less than 1 h.

### 3. Results and Discussion

The electric field intensity distribution in the $x$-$y$ plane is depicted in Figure 2 with $\lambda$ = 660 nm and 799 nm with the polarization angles $\beta$ = 0° and $\alpha$ = 60°. The white dotted curve depicts the quadratic curve $f(y) = -ay^2$. It demonstrates that the target SPBs—both numerous and solitary—are produced with the propagation along the required trajectory. These results demonstrate that SPBs are generated by an X-shaped nanoholes array. A conversion between multiple SPBs and a single one can be realized by changing the incident light wavelength. In addition, with $\lambda$ = 660 nm, the electric field intensities at points A (20 μm, 16.4 μm) and B (20 μm, −4.2 μm) were about 0.03 times of the intensities of incident light wave (magnitude 1 V/m). With $\lambda$ = 799 nm, the electric field intensities at point B (20 μm, −2.7 μm) were about 0.14 times of the intensities of incident light wave (magnitude 1 V/m). The electric field intensity of generated SPB is low. It is mainly related to the structure size, row number of structures, the radiation intensity of SPP wave, and its attenuation in propagation. The findings well confirm our theoretical discussions

of generated SPBs in Equations (5) and (6). The findings could help in designing and optimizing controllable multi-function SPB converters.

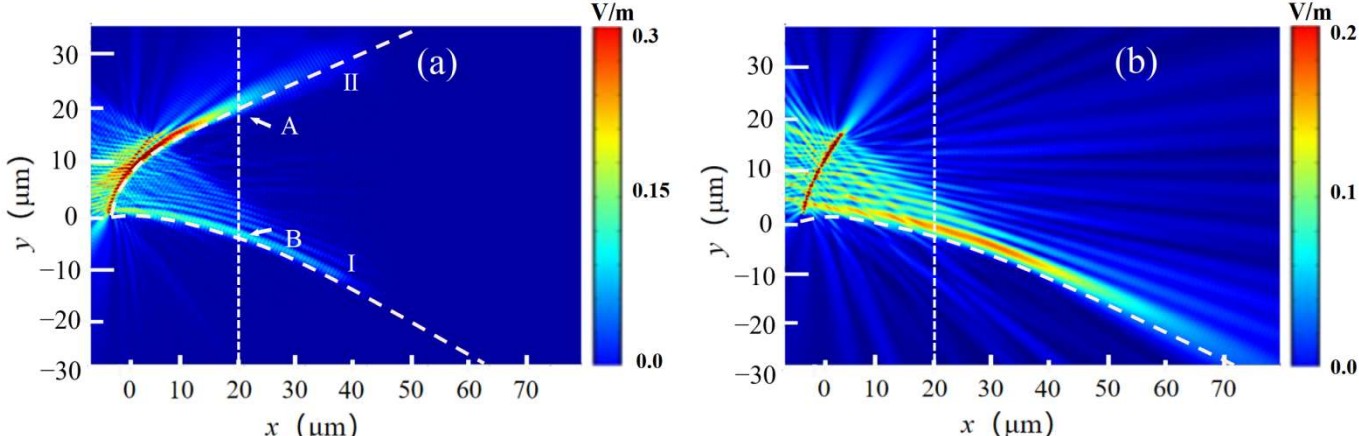

**Figure 2.** The electric field intensity distribution with different wavelengths. (**a**) $\lambda$ = 660 nm, (**b**) $\lambda$ = 799 nm.

The electric field intensity distributions with $\lambda$ = 630 nm, 660 nm, 690 nm, and 710 nm in the *x*-*y* plane with the polarization angles $\beta = 0°$ and $\alpha = 60°$ are shown in Figure 3a. The transverse electric field intensity distributions with different incident light wavelengths $\lambda$ at *x* = 20 μm are shown in Figure 3b. It shows that by increasing $\lambda$ from 630 nm to 710 nm, the electric field intensity of SPB I increases, and the electric field intensity of SPB II first increases and then decreases. The distance $D_n$ represents the vertical distance between SPB I and II in Figure 3c. It shows that with increasing $\lambda$, the distance $D_n$ (corresponding to angle between multiple SPBs) gradually increases. The main reason is that the two SPBs phases ($\varphi_1(x)$ and $\varphi_2(x)$, shown in Equation (6)) change at different rates with respecting to wavelength, resulting in different speeds of two SPBs deviating from the target quadratic curve trajectory. The findings show that the electric field intensities of multiple SPBs are controlled with varying the incident light's wavelength. The angle between multiple SPBs (corresponding to the distance $D_n$) is also controlled by changing the wavelength of incident light.

Figure 4a displays the electric field intensity distributions for $\lambda$ = 850 nm and 980 nm in the *x*-*y* plane with polarization angles $\beta = 0°$ and $\alpha = 60°$, respectively. The transverse electric field intensity distributions with different $\lambda$ (from 750 nm to 1150 nm) at *x* = 20 μm are shown in Figure 4b. These figures show that a single SPB is generated. These results demonstrate that a conversion between multiple SPBs and single one is realized by changing the incident light's wavelength. The conversion point occurs near $\lambda$ = 750 nm. Multiple SPBs are the superposition of SPPs waves from every nanohole, and single SPBs is the superposition of electric dipole radiation from each resonance of X-shaped nanohole. We can follow the literatures [45,53] to qualitatively explain these results. These analyses can be well explained by the Huygens–Fresnel principle in Equation (4). These results demonstrate that a change from SPPs resonance in the nanohole to LSP resonance of an X-shaped nanohole is applied to explain this conversion. In addition, the maximum electric field intensity values of the SPBs with different $\lambda$ values are shown in Figure 4c. Clearly, the electric field intensities of SPBs initially increase with $\lambda$ increasing (from $\lambda$ = 750 nm to 1150 nm), and thereafter decrease.

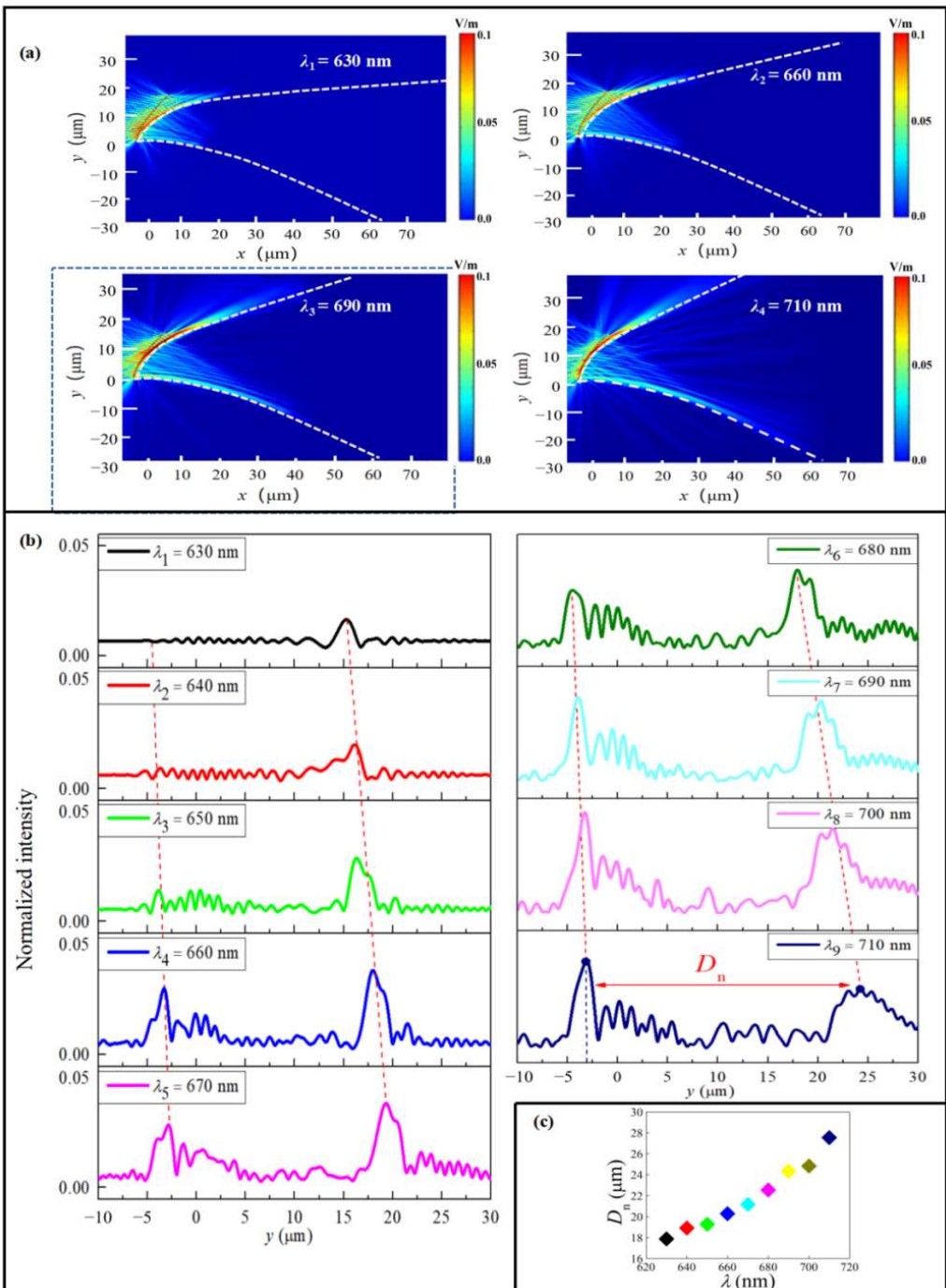

**Figure 3.** The electric field intensity distributions for the proposed structure with different $\lambda$. (**a**) $\lambda$ = 630 nm, 660 nm, 690 nm, and 710 nm, respectively. (**b**) Transverse electric field intensity distributions at $x$ = 20 μm with different $\lambda$. (**c**) The distance $D_n$ with different $\lambda$.

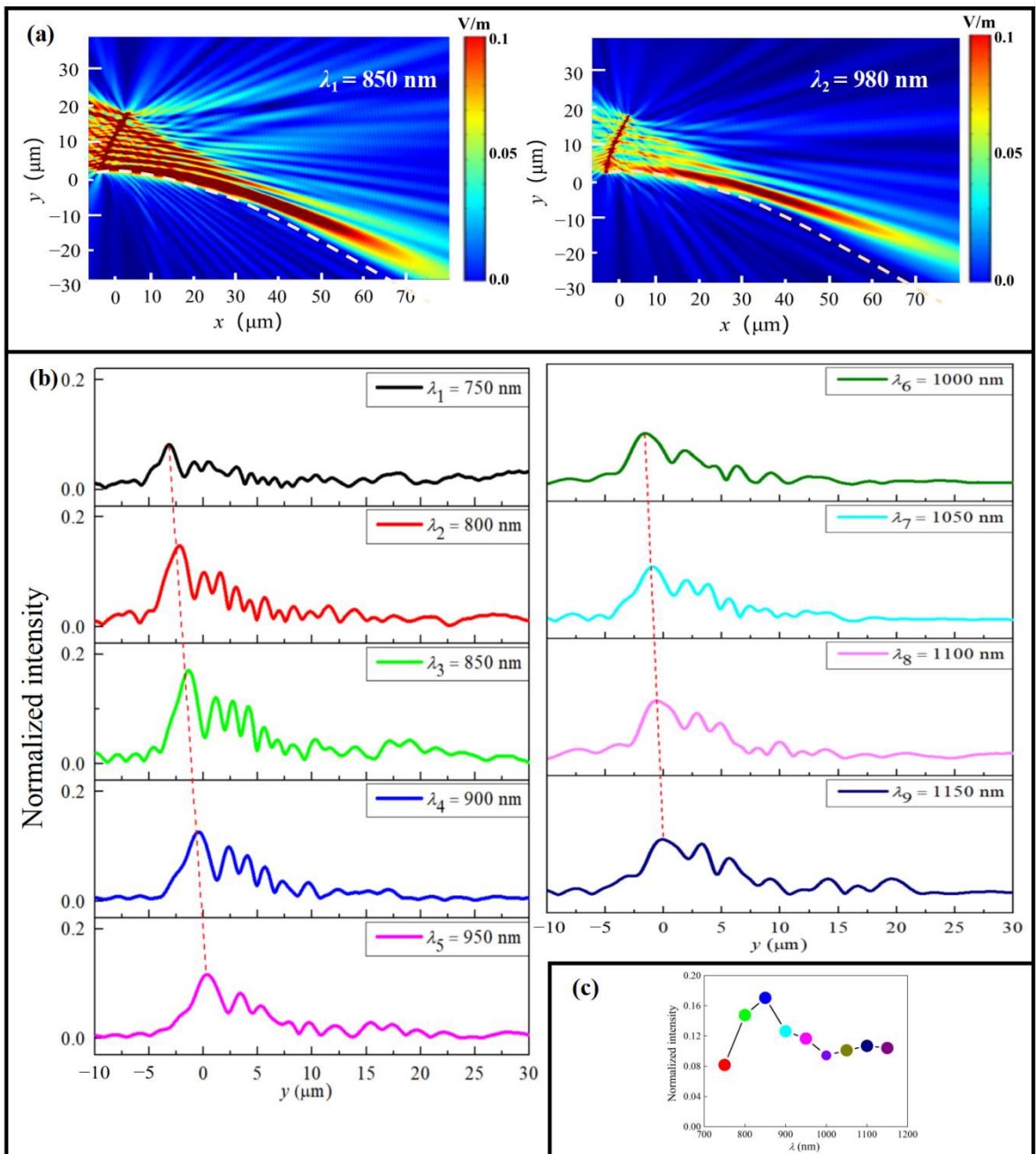

**Figure 4.** The electric field intensity distributions with different $\lambda$. (**a**) $\lambda$ = 850 nm, and 980 nm. (**b**) Transverse electric field intensity distributions at $x$ = 20 μm with different $\lambda$. (**c**) The maximum electric field intensity values of the SPB with different $\lambda$.

To understand the relationship between the electric field intensities of SPBs and the polarization angle $\beta$, the electric field intensities with different polarization angle $\beta$ in $x$-$y$ plane are depicted in Figure 5. Electric field intensities at the selected points A ($x$ = 20 μm, $y$ = 15 + $\Delta D$ μm) of beam II and B ($x$ = 20 μm, $y$ = −4 + $\Delta D$ μm) of beam I are studied to unveil this polarization-dependent property. Here, $\Delta D$ considers the deviation of SPBs. The polar plots of the electric field intensity values of points A and B with varied polarization angle $\beta$ at $\lambda$ = 630 nm, 670 nm, and 710 nm are shown in Figure 5. The equation to fit these data is given by,

$$f = C + D \times \sin(\varphi - \varphi_0),\qquad(7)$$

where $C$ and $D$ are constant. The solid curve in Figure 5 results in the fitting of Equation (7). The goodness of fit for beams I and II are 0.98, 0.96, 0.95, and 0.96, for $\lambda$ = 630 nm, 660 nm, 690 nm, and 710 nm, respectively. These results show the relationship between the electric field intensities of multiple SPBs and the polarization angle $\beta$ satisfies the sine distribution. This analysis is also consistent with our calculation results in Equation (5), satisfying the sine distributions [60].

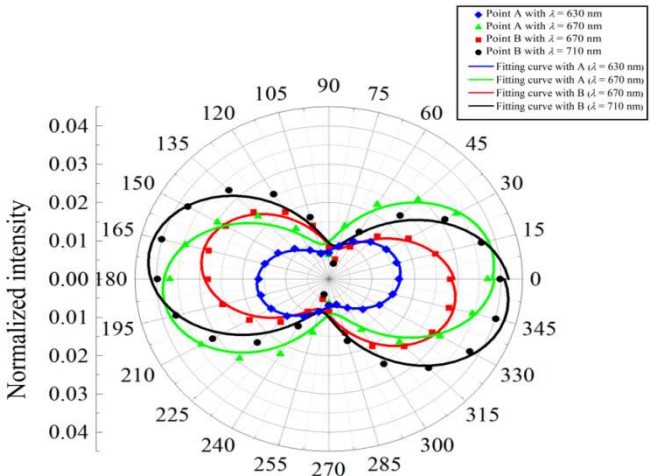

**Figure 5.** The polar diagram for electric field intensities at points A and B with varied polarization angles $\beta$.

It should be emphasized that SPBs can be significantly influenced by the geometrical characteristics of the X-shaped nanohole. The transverse electric field intensity distributions of the X-shaped nanoholes array structure and single nanohole structure for different angles $\theta$ at $x$ = 20 μm (with $\lambda$ = 660 nm, $\beta$ = 0°) are shown in Figure 6a. It shows that the electric intensity values of SPBs for the proposed structure are stronger than that of the single slit antenna. It further demonstrates that multiple SPBs are also generated by single nanohole array. At this time, the electric field intensity distributions of SPBs can be controlled by the angle $\theta$. With the angle $\theta$ increasing from 0° to 180°, the electric intensity values of SPBs first increase and then decrease. The transverse electric field intensity distributions of the proposed structure at $x$ = 20 μm for different widths (with $\lambda$ = 660 nm) are shown in Figure 6b. It shows that as the width $w$ increases from 0.02 μm to 0.07 μm, the electric intensity values of SPBs increase. An increase in the width of the X-shaped nanohole will decrease the confinement of SPPs in the nanohole. Therefore, with an increase in $w$, the effective aperture of X-shaped nanohole increases and electric field intensity of SPPs becomes stronger, and as the length $L$ increases from 0.26 μm to 0.4 μm, the electric intensity values of SPBs first increase and then decrease. Compared with width $w$, the effect of length $L$ on the electric field intensities of SPB is inconspicuous. In addition, the electric field intensities of SPBs are controlled by changing angle $\alpha$ between two arms of X-shaped nanoholes (shown in Figure 6c). This shows that as the angle $\alpha$ increases, the electric field intensities of SPBs increase and then decrease.

In addition, to enrich the conversion results, other triangle (height $h$ = 0.15 μm, width $l$ = 0.2 μm) and circular (radius $r$ = 0.1 μm) nanohole arrays are designed to generate multiple SPBs. The electric field intensities of multiple SPBs are also controlled by changing the incident light wavelength. These results demonstrate that this conversion from multiple SPBs to single SPB can be realized by other structures.

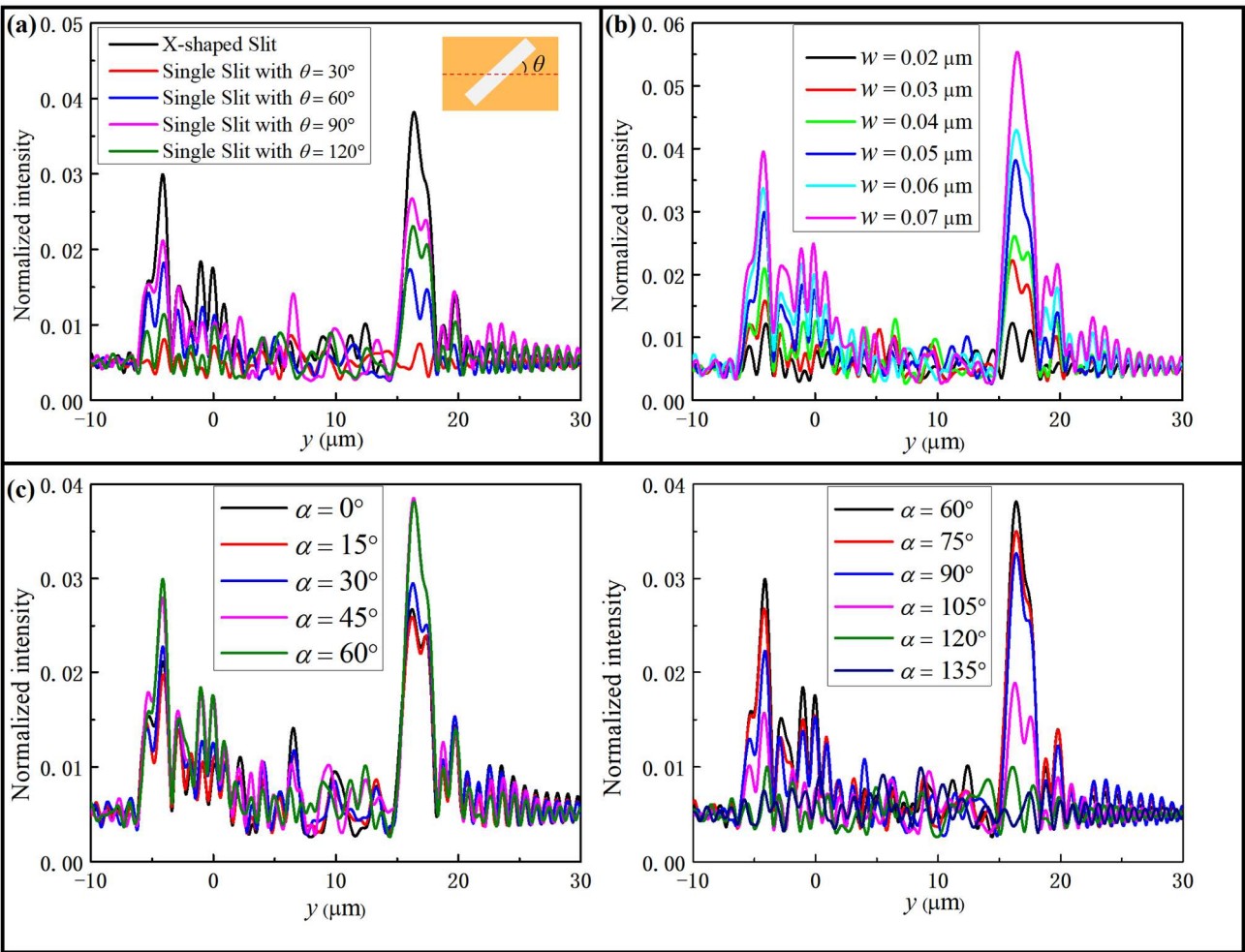

**Figure 6.** (**a**) Transverse electric field intensity distributions for (**a**) single slit with different angle $\theta$, and for the proposed structure with different (**b**) width $w$ and (**c**) angle $\alpha$ with $\lambda$ = 660 nm and $\beta$ = 0°.

The experimental preparation of the proposed structures is feasible. For the fabrication of the proposed structures, an Electron Beam Lithography (EBL) system and a Vacuum Evaporation Coating (VEC) machine will be applied. S1: 250-nm-thick PMMA (as negative photoresist) is dumped on the SiO$_2$ substrate. The substrate after dumping is dried on a hot plate at 150 °C for 3 min. Conductive adhesive with 20 nm thick is dumped on the PMMA to get sample 1. S2: Sample 1 is processed by EBL system, cleaned in deionized water, washed out conductive adhesive, and then soaked in development solution and finalization solution for 60 s, then dried to get sample 2. S3: 200-nm-thick Au film is vertically plated using Vacuum Evaporation Coating Machine to get sample 3. S4: Sample 3 is soaked in acetone solution for 3h, then peeled off and dried to get the proposed structures.

## 4. Conclusions

In conclusion, an X-shaped nanohole array was designed to generate multiple SPBs. The angle between multiple SPBs is dynamically controlled by changing the incident light wavelength. A conversion between multiple SPBs and a single one is realized by only changing the incident light wavelength. The relationship between the electric field intensities of the SPBs and the polarization angle of the incident light satisfies the sine distribution, and is consistent with theoretical analysis of dipole radiation. The electric field intensities of SPBs rely on the structural width and length. These findings could help in the design and optimization of controllable bending beam generators.

**Author Contributions:** Conceptualization, L.W., H.Z. and H.L.; methodology, L.W., H.Z. and H.L.; software, L.W.; validation, L.W., H.Z. and X.L. (Xueli Li); formal analysis, L.W.; investigation, X.L. (Xiaoming Li); resources, H.Z.; data curation, X.L. (Xueli Li); writing—original draft preparation, L.W.; writing—review and editing, L.W. and H.Z.; visualization, L.W. and H.Z.; supervision, L.W. and H.Z.; project administration, L.W. and H.Z.; funding acquisition, H.L. The authors L.W. and H.Z. contributed equally to this work. All authors have read and agreed to the published version of the manuscript.

**Funding:** This research was funded by the following projects: the Yunnan Local Colleges Applied Basic Research Projects (no. 2019FH001(-081)), the Science Foundation of Yunnan Provincial Education Department (no. 2020J0514, 2022Y720, 2022Y727), the Talents Introduction Project of Kunming University (YJL2006), and the Frontier Research Team of Kunming University 2023 (XJ20230042).

**Institutional Review Board Statement:** Not applicable.

**Informed Consent Statement:** Not applicable.

**Data Availability Statement:** Not applicable.

**Conflicts of Interest:** The authors declare no conflict of interest.

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
