# Peer review of "Tunable Multiple Surface Plasmonic Bending Beams into Single One by Changing Incident Light Wavelength"

_photonics, doi:10.3390/photonics10070758_

Round 1
Reviewer 1 Report
In this paper, the authors proposed a multifunctional metasurface that can generate different SPBs by changing the incident wavelengths. The simulated results indicate that tunable multiple SPBs into single one can be achieved. There are some issues that should be addressed before this paper can be accepted for publication.
1. The language of this paper should be further improved, which can make this paper more like a journal paper.
2. The detailed information of the proposed structure should be further discussed. For example, the period of the unit cell or the distance between the unit cells, how many unit cells are designed.
3. The simulation settings should be further explained so that the readers can reproduce this work.
4. The working efficiency of the device is quite low that the authors need to explain the reasons.
5. The authors should further explain how to implement desired phase response by the proposed unit cell.
6. I suggest the authors adding theoretical trajectories of the beams in corresponding simulated images to demonstrate the validation of the design method.
7. When considering metasurfaces for self-accelerating beam generation, the following references may be helpful. 10.1002/adom.201900503, Nano Lett. 2018, 19, 1158, Nanophotonics 2020, 9, 20200057, and Annalen der Physik, 2020, 532(1): 1900396.
Minor editing of English language required
Reviewer 2 Report
The authors theoretically studied on how to tune multiple surface plasmonic bending beams into single one by changing incident light wavelength. This work is interesting and has potential applications in fiber sensor, optical trapping, and micro/nano manipulation of light. I would like to recommend this manuscript for publication, after the following concerns are addressed properly.
[1] The English of this manuscript need to be carefully checked:
In Line 18, “satisfy” should be “satisfies”.
In Line 19, “theoretical” should be “theory”.
In Line 25, “a” should be deleted.
In Line 30, “it has” should be “they have”.
In Line 40, “is” should be “it is”.
In Line 52, “which” should be “which is”.
In Line 62, “separate” should be “separating”.
In Line 73, “to” should be deleted.
In Line 77, “satisfy” should be “satisfies”.
In Line 77, “consistent” should be “and is consistent”.
In Line 92, “λis” should be “λ is”.
In Line 123, “are” should be “is”.
In Line 129, “0.3um” should be “0.3 um”.
In Line 130, “0.05um” should be “0.05 um”.
In line 160, “intensities” should be “intensity”.
In line 186, “distribution” can be deleted.
In Line 243, “consistent” should be “and is consistent”.
[2] In this work, no experimental data were provided to confirm the numerical results. Therefore, for the presented results to be reproduced well, I strongly suggest the authors to further describe the details of numerical calculations performed by the Lumerical FDTD solutions software.
[3] There are very relevant works about plasmonic refractive index sensors: Results in Physics 47, 106354 (2023); Journal of Optics 25(1), 015001 (2023). These papers can be cited to give audience a broader picture of this field.
[4] The authors demonstrate that a conversion between multiple SPBs and single one is realized by changing incident light wavelength. However, the underlined physical reasons are not discussed in detail, which should be added in the revised manuscript.
Reviewer 3 Report
The authors proposed multiple surface plasmonic bending beams into single one by changing wavelength. Although this manuscript is interesting, there are many problems, such as English problems, that need to be carefully addressed:
1. The authors have used the CSC model for Au refractive index for their simulations, which is not as realistic a model as the Lorentz-Drude model. Why did the authors not use the Lorentz-Drude model? The authors are advised to explain this issue in their manuscript.
2. Abstract is poorly written.
3. The English are poor in this manuscript; there are many English problems.
4. The fabrication process of the structure should explain by the authors.
5. In regards to the plasmonic structures, the authors should mention other configurations in the introduction section such as:
¾ High sensitivity plasmonic refractive index sensing and its application for human blood group identification. Sensors and Actuators B: Chemical, 249, 168-176, 2017.
¾ Gold Nanoparticle-Based Plasmonic Biosensors. Biosensors, 13(3), 411, 2023
¾ High-sensitivity plasmonic sensor based on metal–insulator–metal waveguide and hexagonal-ring cavity. IEEE Sensors Journal, 16(9), 3041-3046, 2016.
¾ Design of a plasmonic sensor based on a square array of nanorods and two slot cavities with a high figure of merit for glucose concentration monitoring. Applied optics, 57(27), 7798-7804, 2018.
6. Authors should compare their results with other articles that have been published on the topic of plasmonic bending in a table.
7. Can the authors compare the output power vs. input power in Fig. 1. For example, for a wavelength of 660nm. Actually, they should calculate the amount of attenuation of the input light signal.
8. Authors are required to compare their results with the case of a straight bent waveguide.
The English are poor in this manuscript; there are many English problems.
Round 2
Reviewer 1 Report
I'm glad that the revised version has solved my previous concerns and the manuscript quality has been greatly improved. I suggest it can be accepted now.
Reviewer 2 Report
All my questions have been addressed properly, and this work can be accepted for publication.